# Aggressive Angiomyxoma of the Lower Female Genital Tract in Pregnancy: A Review of the MITO Rare Tumors Group

**DOI:** 10.3390/cancers15133403

**Published:** 2023-06-29

**Authors:** Stefania Cicogna, Miriam Dellino, Salvatora Tindara Miano, Francescapaola Magazzino, Lavinia Domenici, Sandro Pignata, Giorgia Mangili, Gennaro Cormio

**Affiliations:** 1Institute for Maternal and Child Health-IRCCS “Burlo Garofolo”, 34145 Trieste, Italy; stefania.cicogna@gmail.com; 2Department of Interdisciplinary Medicine (DIM), University of Bari “Aldo Moro”, Piazza Giulio Cesare 11, 70124 Bari, Italy; gennaro.cormio@uniba.it; 3Complex Operating Unit of Oncology, Azienda Ospedaliera Universitaria Senese, 53100 Siena, Italy; doramiano@hotmail.com; 4Complex Operating Unit Ginecologia E Ostetricia, Ospedale Civile Di San Dona’ Di Piave (Venezia), Aulss4 Veneto Orientale, 30027 San Donà di Piave, Italy; framagazzino@hotmail.it; 5Division of Obstetrics and Gynecology, Azienda Ospedaliera Universitaria Pisana, University of Pisa, 56126 Pisa, Italy; lavinia.domenici@gmail.com; 6Department of Urology and Gynecology, Istituto Nazionale Tumori IRCCS ‘Fondazione G Pascale’, 80144 Napoli, Italy; s.pignata@istitutotumori.na.it; 7Obstetrics and Gynecology Unit, IRCCS San Raffaele Scientific Institute, 20132 Milan, Italy; mangili.giorgia@hsr.it; 8Gynecologic Oncology, IRCCS Istituto Tumori “Giovanni Paolo II”, 7012 Bari, Italy

**Keywords:** aggressive angiomyxoma, deep angiomyxoma, vulva, pregnancy

## Abstract

**Simple Summary:**

Deep (aggressive) angiomyxoma (DAM) is an extremely rare primary neoplasm of the vulva whose growth is particularly favored during pregnancy. To date, neither the origin nor the optimal management of DAM are clear, especially in pregnancy. Therefore, we present here a systematic review that could be a useful tool to provide further knowledge about the behavior of and the management strategies for this rare malignancy.

**Abstract:**

Deep (aggressive) angiomyxoma of the lower genital tract is a rare malignancy affecting women of reproductive age. Being a hormone-sensitive tumor, its growth is particularly benefitted during pregnancy. Surgical excision with complete resection is indicated, even if a wait-and-see approach can be considered until delivery, to avoid destructive surgeries. The mode of delivery is to be evaluated based on the location and size of the neoplasm; vaginal delivery is not contraindicated, as long as the tumor does not obstruct the birth canal. Positive surgical margins are the most important prognostic factor for recurrence. Adjuvant therapy with gonadotropin-releasing hormone analogues may be proposed after pregnancy, in the case of non-radical surgery. Despite the high local relapse rate, the outcomes for mother and child are favorable. Since recurrences can occur after many years, the patient should be included in long-term follow-up.

## 1. Introduction

Deep (aggressive) angiomyxoma (DAM) is a rare mesenchymal tumor that arises in the deep soft tissue of the vulvovaginal region, perineum, and pelvis. It is a neoplasm that occurs mainly in women of reproductive age, with a median age between 30 and 40 years [1,2]. Since it derives from hormone-sensitive stromal tissues of the lower genital tract, the progression of this neoplasm is particularly favored during pregnancy. According to the latest WHO classification, DAM is considered a benign tumor, but it is defined as “aggressive” due to its ability to infiltrate adjacent structures and its capacity to recur, especially locally [1]. From 25% to 47% of recurrences are described after resection, even with negative margins and 10–15 years after primary excision [3]. In pregnant women, DAM typically presents as an indolent polypoidal mass, frequently with consistent dimensions, arising from the genitalia or perineum. Due to its appearance, DAM is often misdiagnosed as other more frequent conditions, such as a Bartholin’s cyst, lipoma, labial cyst, Gartner’s duct cyst, condyloma acuminata, vaginal prolapse, or an unspecified tumor [4,5,6,7,8,9,10,11,12]. The diagnosis is performed by the pathologist and not by the clinician, thanks to immunohistochemistry, which is essential to distinguish DAM from other benign stromal tumors of the lower genital tract. After the first report of DAM of the vulva, described by Steeper and Rosai in 1983 [13], the first recorded incidence of DAM in pregnancy was published by Fishman and Otey in 1995. Only a few cases diagnosed in pregnancy, which are described below, have been reported in the literature. Given the rarity of this neoplasm, it can be difficult to identify the best course of management, especially if the diagnosis is made during pregnancy. The aim of this research is to carry out a systematic review of the literature in order to describe the clinical and prognostic characteristics of DAM in pregnancy and to propose tailored treatments appropriate to the gestational age at diagnosis, balancing the risks and benefits for both mother and child.

## 2. Materials and Methods

A systematic review of DAM case reports was performed through a literature search of the following electronic databases: PubMed, the Cochrane Library, Embase, the Web of Science, and Medline. The research was performed in agreement with the Preferred Reporting Items for Systematic Reviews and Meta-Analyses (PRISMA, Figure 1) [14]. The following search terms were used: “aggressive angiomyxoma”, “deep angiomyxoma”, “vulvar” “pregnancy”. No restrictions regarding the publication period were applied. Particularly, we considered case series and case reports published in English. Titles and abstracts of the eligible articles were independently reviewed by two authors (S.C and M.D). Duplicates were removed. The full texts of potentially suitable studies were independently assessed for eligibility by the two authors. Any discordance between the two authors were solved through discussion with two senior reviewers (G.C. and G.M.). Data were retrieved from articles published from 1995 (in which Fishman first described a case of aggressive angiomyxoma of the vulva in a pregnant woman) until December 2022. Articles reporting DAM in non-pregnant women were excluded.

## 3. Results

From the literature search, we identified 17 articles, reporting a total of 19 cases, which are listed in the Table 1.

### 3.1. Etiopathogenesis

The risk factors for DAM are still unknown. No exogenous risk factors have yet been identified as a direct carcinogen for the development of DAM; however, some genetic rearrangements have been identified. In the literature, a translocation to chromosome 12 is reported to be present in one-third of the cases of DAM, resulting in the aberrant expression of high mobility group A (HMGA) protein involved in DNA transcription [27,28]; in particular, rearrangements of the HMGA2 (12q14.3) locus have been identified in 8 chromosomal translocations involving HMG genes, which have been previously reported in various other mesenchymal neoplasms, including lipomas, liposarcomas, leiomyomas, and pulmonary hamartomas (Figure 2) [3,29,30]. RT-PCR studies have confirmed the presence of aberrant HMGA2 transcript expression in deep angiomyxomas exhibiting HMGA2 locus rearrangement [31,32]. HMGA protein is furthermore identifiable by immunohistochemistry in neoplastic stromal cells; its use has been proposed as a diagnostic tool in assessing surgical margins status, which could be extremely difficult to evaluate by macroscopic analysis alone [33]. However, in none of the cases described in pregnancy was this protein tested by immunohistochemistry, nor was a molecular investigation performed, likely due to the small number of cases diagnosed during pregnancy. In all cases of DAM, estrogen and/or progesterone receptors were identified by immunohistochemical analysis. The presence of hormone receptors justifies the diagnosis, as well as the recurrence, of this rare mesenchymal neoplasm during the period of pregnancy and puerperium, supporting a strict hormonal dependency.

### 3.2. Clinical Features

DAM in pregnancy is often misdiagnosed as other more common vulvar heteroplasia, such as a Bartholin’s gland cyst, lipoma, or a vaginal wall cyst [4,5,6,7,8]. The clinical presentation of deep angiomyxoma depends on its size and localization, but typically, at the onset, it presents as an indolent mass arising from the genitalia and the pelvic-perineal region, usually in the first or second trimester of pregnancy, with a tendency to grow rapidly during gestation. In some cases, patients have reported a previously removed vulvar neoplasm of unknown histology, or that they were aware of having a known vulvar formation during the pre-pregnancy period, which increased during pregnancy [8,18]. DAM is most frequently identified at the end of the first trimester and during the second trimester of pregnancy, but some cases have also been identified in the third trimester or during the postpartum period [3,17].

Since these are mesenchymal tumors without a defined capsule, DAMs tend to infiltrate deep into the surrounding tissues, in some cases reaching the deep pelvis. These tumors are often much larger and extend deeper than is initially noticeable during pelvic examination. Other than the presence of a vulvovaginal mass or swelling, no other symptoms, such as bleeding or pain, are reported. No cases of metastatic onset during pregnancy or distant recurrences following pregnancy have been reported, indicative of an excellent prognosis. In addition to the clinical examination, it is useful to perform instrumental tests to better define the extent of the disease and to determine the appropriate therapeutic procedure. Transvaginal ultrasound is often used as the first diagnostic tool, identifying a solid neoformation with a hyper-isoechoic echostructure with indefinite margins, richly vascularized using color Doppler [17,24]. Given the location of deep angiomyxoma and its ability to infiltrate the perineal and pelvic structures, the most appropriate imaging technique is magnetic resonance imaging (MRI). Characteristic appearances on magnetic resonance imaging include hypointensity on T1-weighted images and hyperintensity on T2-weighted images [6,17,24,26]. DAM exhibits heterogenous enhancement after administration of intravenous contrast [3]. Diffusion-weighted imaging shows uneven, slightly high signal shadows [25]. In some older cases (published more than 10 years ago), computed tomography (CT) was used in the postpartum period, but it never highlighting distant locations [17,18,20].

### 3.3. Pathological Examination

On overall examination, deep angiomyxoma in pregnancy mostly presents as a large, lobulated or polypoid soft mass removed from the vulvar-perineal region with ill-defined, often positive, surgical margins [6,18,19,23,26]. The diameter ranges from a few centimeters up to 55 cm, with most measuring > 5 cm at first diagnosis [17,19,20,21,23,24,25,27]. When cut, upon macroscopic examination, the tumor appears translucent, gray-white or gray-brown, exhibiting a homogeneous gelatinous consistency, with cystic changes and bleeding areas [25]. The tumors lack a capsule, which is the reason for the infiltrative pattern of this neoplasm [1]. The microscopic appearance of DAM is that of an ill-defined, hypocellular, infiltrative lesion composed of spindle or stellate cells scattered among abundant edematous myxoid stroma. There is usually a prominent vascular component, characterized by thick-walled vessel [34]. Neoplastic cells exhibit a poorly defined, scant cytoplasm and round, hyperchromatic nuclei; mitoses are absent or rare [35]. By immunohistochemistry, neoplastic spindle cells stain positively for vimentin, desmin, and smooth muscle actin (SMA) [35]. Variable expression of CD34 is detected; S100 and cytokeratin markers are usually negative [9]. The Ki-67 proliferation index is consistently low (<1% of tumor cells) [25,26]. ER and PR receptors are commonly positive in the stromal cells of DAM [36], and the data was confirmed for the cases analyzed; the relevant hormonal changes in pregnancy therefore justify the abnormal growth, as well as the high rate of recurrence, of these neoplasms during pregnancy and the postpartum period. HMGA2 nuclear immunoreactivity, even if not specific, is present in most cases of DAM and could be useful in distinguishing it from other mesenchymal tumors [37]; however, as mentioned above, this was not tested in any of the cases occurring in pregnancy. DAM is considered a benign tumor; however, recurrence is reported to be 30–40%, and sometimes even higher, up to 50% [2,38]. This high local-recurrence rate was confirmed by the 19 cases analyzed during pregnancy, of which four relapsed and two were considered suspicious for recurrence [6,8,15,18,19,25]; in all of the recurrent cases, if the data were reported, the margins of the surgical specimen were positive.

### 3.4. Management

The first choice of treatment for DAM is surgery, with a local wide excision, with the goal of achieving free edges, although attaining negative resection margins could be difficult because of the infiltrative nature of the tumor and the absence of a defined capsule [3]. Complete excision may involve the removal of adjacent structures, such as fascia, muscles, and neighboring organs, constituting potentially destructive surgery. As a hormone-sensitive neoplasm, DAMs diagnosed during pregnancy, unlike those identified in non-pregnant women, were treated conservatively until pregnancy reached full term, or surgically in the second or third trimester of gestation, with comparable outcomes for mothers and newborns. Of the 19 patients noted in our review, cases which were highly symptomatic or suspicious for neoplasia tended to be surgically treated during pregnancy; in most of these patients, after the surgery, the pregnancy proceeded without complications and the patients were able to achieve a vaginal delivery at term [4,5,7,8,16,21]. Patients with paucisymptomatic DAM were followed up until the end of the pregnancy; in some cases, delivery was accomplished by Cesarean section due to the presence of the tumor obstructing the birth canal [17,20,22,23,26]; surgical excision was then postponed after the delivery. In some cases, spontaneous regression after pregnancy is reported. Given the rich vascularization, in case of surgical excision, the potentially high risk of heavy bleeding should be taken into account [37]. In the case of suboptimal surgery with positive margins, in order to reduce the complications of a destructive surgery, adjuvant therapy was sometimes proposed after delivery. In particular, adjuvant therapy with gonadotropin-releasing hormone (GnRH) analogues has been suggested for 6–9 months after surgery [6,19,25,26]. In our cases retrieval, adjuvant radiotherapy was performed in only one patient (60 Gy) because the patient presented positive margins, with no evidence of recurrence for 8 years [18]. However, there is no evidence in the literature that adjuvant radiotherapy will lower the recurrence rate. In non-pregnant cases, as an alternative to demolitive surgery, angiographic embolization has been proposed to reduce the mass by devascularizing the tumor through the hypogastric artery. However, recurrence after angiographic treatment is reported, possibly due to the development of an alternative blood supply to the tumor. For this reason, embolization should be reserved as a second choice for recurrent cases not amenable to surgery and/or unresponsive to other treatments. The local recurrence rate for DAM is about 30–40% [2], which was also confirmed by our data on pregnant patients (in which 4 out of 19 patients relapsed, and another two probably presented relapsed diseases at the first diagnosis); as mentioned above, all of these patients had positive margins at surgical resection. Recurrence can occur, even after many years (in our cases, up to 8 years later. Therefore, long-term follow-up with clinical examination and MRI is suggested, even in cases where no unfavorable outcome, despite the recurrence, is reported.

## 4. Discussion

Deep (aggressive) angiomyxoma is a slow-growing, benign, but locally infiltrating neoplasm with a high local-recurrence rate, even many years after excision. DAM can reach considerable dimensions during pregnancy, probably due to its hormonal dependence, given the presence of the estrogen and/or progesterone receptors. Since the diagnosis is histological, in most cases, DAM is misdiagnosed as other more frequent vulvar neoplasm, such as a vulvar cyst or lipoma. In most cases occurring in pregnancy, it is first identified as a vulvovaginal mass or swelling in the first or second trimester of pregnancy and is either completely excised or biopsied, after which the correct diagnosis is made. In some patients, the tumor identified before or during pregnancy grew abnormally postpartum. Due to its high potential for local recurrence, it is important to distinguish DAM from the other stromal tumors of the female pelvis, including cellular angiofibroma, angiomyofibroblastoma, and myofibroblastoma, focusing on the immunohistochemical features [35]. The treatment of choice for DAM is surgery with local excision of the tumor, aiming for complete resection with negative margins. However, due to the size and infiltration of the neoplasm, it is not always possible to achieve a complete surgical radicality without subjecting the patient to destructive surgery. On the basis of the cases collected in this review, however, it is possible to state that conservative management, delaying treatment until the end of pregnancy, is equally feasible and safe, postponing surgery until after delivery. The therapeutic strategy to be pursued is strictly connected to the size and localization of the neoplasm. If DAM is paucisymptomatic, or if the infiltration of the tumor into the surrounding tissues implies a destructive surgery, it is feasible to opt for a conservative approach until delivery. If the tumor is highly symptomatic, creating discomfort for the patient, and its size allows a local excision, surgical removal is practicable from the second trimester onwards. Among the 19 patients garnered from our review, 10 cases were treated surgically during pregnancy; in most of these patients, after surgery, the pregnancy continued without complications and they were able to achieve a vaginal delivery at term [4,5,7,8,16,18,21,24]. Conversely, for patients treated with conservative or non-radical management, delivery was accomplished by Cesarean section, in most cases, due to the presence of the tumor obstructing the birth canal [6,17,22,23,25], postponing the surgical excision until after delivery. Delaying surgery that may potentially be non-radical or destructive until after delivery is quite safe, especially because in some cases, spontaneous regression of the tumor after pregnancy has been reported [25,39,40,41]. Although complete resections is desirable, reaching negative surgical margins during pregnancy is not mandatory, especially if this results in destructive surgery, as the outcomes are favorable in either case. None of the cases reported in pregnancy, whether treated conservatively or surgically, recurred with distant metastases or resulted in death. Distant metastatic disease is extremely rare. In the literature, only two cases with distant metastases (lung, peritoneal, and lymph node metastases) are described [38,42]; neither case was found in pregnancy. Positive surgical margins are a poor prognostic factor for relapse (indeed, all reported recurrent cases exhibited positive margins), and they do not impact survival. In general, deep aggressive angiomyxoma is associated with good maternal and infant outcomes. Cesarean section is not the preferred method of delivery, but it is indicated if the tumor prevents vaginal delivery. Newborns of pregnant mothers diagnosed with DAM were all delivered at greater than 37 weeks of gestational age. In the case of suboptimal surgery with positive margins, adjuvant therapy can be proposed after pregnancy. It is advisable to wait a few weeks after the postpartum period, since there is a significant hormonal drop during this period, and a spontaneous regression of the disease can occur. In case of persistence of positive margins, adjuvant therapy with GnRH analogues (e.g., triptorelin) is recommended until clinical negativity and for at least 6–9 months after surgery. The use of an aromatase inhibitor (letrozole) or a selective estrogen receptor modulator (SERM—e.g., tamoxifen, raloxifen) has been proposed as a second-line hormonal therapy in non-pregnant patients [10,11,12,38]. The role of radiotherapy is limited, and there is no evidence in the literature that adjuvant radiotherapy lowers the recurrence rate. Since these are tumors with a low mitotic index, they may exhibit a limited response to radiotherapy [18]. In non-pregnant cases, tumor embolization by selective angiography has been proposed [18], but given the absence of a central vessel and the possibility of recurrence due to the collateral vascularization, embolization should be reserved as a secondary choice for non-pregnant recurrent cases which are not suitable for surgery and/or are unresponsive to other treatments. For recurrences, where possible, it is always better to perform radical surgery. Patients should be aware of the possibility of recurrence, even after a complete resection. Since recurrences can occur many years after surgery, long-term follow-up is necessary for these patients. A scheduled follow-up consisting of clinical examination and MRI is necessary to detect early relapses. Considering the absence of metastases among the cases diagnosed during pregnancy and the subsequent recurrences, we can assume that it is not necessary to perform computed tomography (CT) as a further instrumental investigation. The strength of our study is that it the largest systematic literature review to date of cases of deep angiomyxoma in pregnancy; it is also the first study in which therapeutic strategies are outlined in relation to clinical presentation and gestational age. A limitation of our study is certainly the small sample size and the fact that the data are not always complete, especially with regards to follow-up. New publications in the literature are needed in order to document further aspects of this neoplasm, especially in relation to the new molecular diagnostic techniques. The aim of the study was to shed light on an extremely rare neoplasm, which has been poorly described and frequently misdiagnosed. It is important for clinicians not to underestimate a new mass arising or grown from the genitalia during pregnancy and to determine whether it could be a rarer and more aggressive disease rather than a more frequently occurring benign tumor.

## 5. Conclusions

The management of DAM in pregnancy can be complex. Local control with surgery alone can be challenging, potentially exposing the patient to destructive surgery. Treatment of this neoplasm in pregnancy should be tailored to the specific case, after balancing the risks and benefits for both mother and child. It is necessary to carefully evaluate the clinical characteristics of the tumor, the local extension, and the gestational age at onset during pregnancy in order to plan the most appropriate treatment and mode of delivery. Surgical excision with complete resection is preferable, when practicable, after the first trimester, but a wait-and-see strategy until delivery occurs can be proposed as equally safe. Although positive margins are a poor prognostic factor for relapse, they do not impact survival in pregnancy. DAM in pregnancy presents a favorable outcome; despite this, further data on the follow-up of these patients are needed to better define long-term therapeutic strategies.

## Figures and Tables

**Figure 1 cancers-15-03403-f001:**
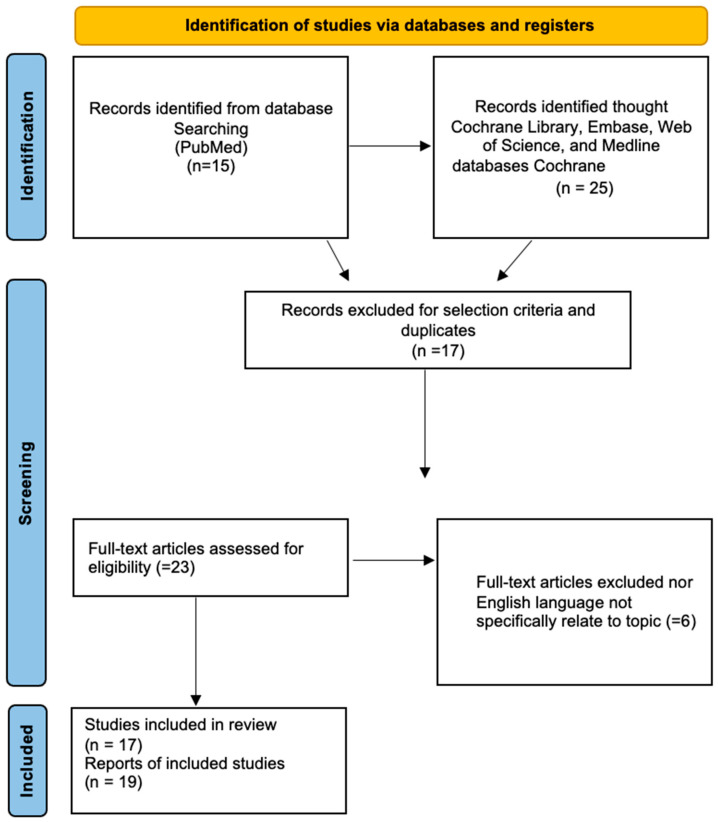
Study flow schema: PRISMA flow diagram of the process of identification, screening, and inclusion of articles. Systematic literature reviews were selected, using standard methods to be briefly presented in the article.

**Figure 2 cancers-15-03403-f002:**
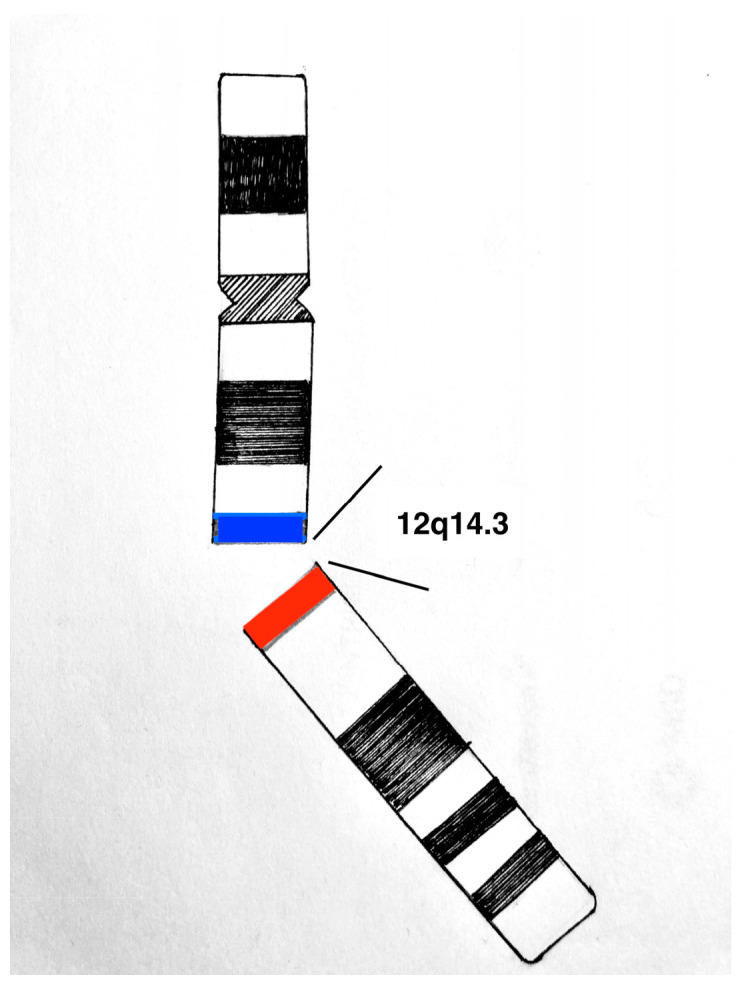
Schematic representation of a possible rearrangement of the HMGA2 locus on chromosome 12.

**Table 1 cancers-15-03403-t001:** Overview of cases of Aggressive Angiomyxoma of the lower female genital tract in pregnancy in the literature revisited until December 2022.

Reference	Age (Years)	GA at Diagnosis (Weeks)	Location	Tumor Size (cm)	IHC	Treatment	Delivery (GA Weeks)	Recurrence	PFS (Months)
[15]	37	NA	Right vulva	3 → 40	NA	Pre-pregnancy LE	NA	Yes	NA
[4]	41	18	Left labium minus	6 × 6 × 4	PR+++, ER+, S100−	LE at 18 weeks	VB (40)	NA	NA
[16]	32	32	Vulva posterior commissura	3 × 4	ER+, PR−	LE 36 weeks	VB	No (9 months follow-up)	NA
[17]	36	Present before and during the pregnancy, grew very rapidly after the birth.	Right major labium, pelvic-perineal region (between the obturator and levator muscles of the anus)	6.5 ≥ 15	Vimentin++ SM actin++ desmin++ ER++ PR++ S100− CD34−	LE postpartum + transperineal surgery (ischio-rectal fossa toilette + external sphincterotomy and reconstruction) (R0)	CS for breech presentation (40)	NA	NA
[18]	31	NA (already present)	paravaginal/pararectal mass	NA	NA	Posterior exenteration (R1) and RT 60 Gy	NA	No recurrence after surgery (first suspected during 1st pregnancy) during second pregnancy 4 years later	96
[18]	34	30	Left and right labia majora	NA	NA	left labial surgery (30 weeks GA; several weeks later, right labial mass surgery was performed); vulvectomy (R1)	NA	No	24
[18]	27	Early pregnancy	Pelvis-perineum, in front of the bladder	NA	NA	Total exenteration (R1)	NA	No	48
[5]	25	12	Right labia majus	2 → 4	Desmin+ ER+ PR−	LE 16 weeks	VB (40)	No	9
[6]	22	Before pregnancy, recurrence at I trimester	Right vulva and pelvis	8 (first diagnosis, before pregnancy) 3.1 × 1.9 × 2.2 (I trimester) → 5.1 × 4.6 × 3.5 (32 weeks), halved in size postpartum	NA	prenatal surgery (R1) + GnRH analogous + surgery + GnRH monitored in pregnancy	CS (38) (for mass)	Yes (in pregnancy)	NA
[19]	22	Before pregnancy, recurred three times (last one during pregnancy)	Right vulva and vagina extending into right ichiorectal-fossa and levator ani	NA	NA	3 LWE → successful pregnancy after third surgery (R1) + GnRH analogue	NA	Yes (before and during pregnancy)	5
[20]	43	tumour gradually increased for 9 years, suddenly grew during and after pregnancy	Left labium majus	Up to 55 (postpartum)	ER+	LE 9 months postpartum	CS (NA)	No	8
[7]	25	18	Left labium majus	Up to 8	NA	LE 18 weeks	VB (40)	No	9
[21]	24	16	Right labium majus	Up to 30	ER+ PR+ desmin+	LE 20 weeks (R0)	CS for failed labor induction (40)	No	60
[22]	21	20	Right labium majus	Up to 15 in pregnancy, up to 18 postpartum	NA	LE postpartum (6 weeks) (R0)	CS for vaginal mass (38)	No	NA
[8]	25	9	Vaginal fornix	12	Vimentin+ SMA+ ER+ PR+ s100−, EMA−, CD34−	LE 13 weeks	VB (40)	No, after surgery (suspected REL in pregnancy)	
[23]	29	20	Right labium majus	2 → 7	ER+ PR+ CD34+	LE during CS (R1)	CS for vaginal mass (39)	No	20
[24]	24	17	Vaginal fornix (bleeding, pain, vaginal mass)	11.4 × 11.3 × 9.95	ER+ vimentin+ PR− S100−	LE during abortion	Induced abortion	NA	NA
[25]	32	8 (2 months)	Right vulva	3 → 5 (first pregnancy), up to 7 cm postpartum	Vimentin+++ ER++ PR++ desmin+ Ki-67 + (<1%) S100−	No treatment	VB (37)	Yes 40 months (during second pregnancy)	8
						LE 22 postpartum			

				6 × 4 × 3 (second pregnancy)		spontaneuos regression postpartum	CS for fetal distress (39)	8 months after the second pregnancy, postpartum	
				5.3 × 5.1 × 4.2		LE 11 months postpartum +GnRH analogous			
[26]	36	20	Prolapsed vaginal mass, vagina, vesico-vaginal space	7 × 3 × 3.5 (enlarged in puerperium 7 × 7 × 7.5)	Vimentin+, Actin+/− CD34+/− S-100−, Desmin− ER+ PR+ EMA− Ki 67 < 1%	LE postpartum (R1) + GnRh analogues	CS for prolapsed mass (37)	NA	NA

Legend: GA = gestational age; IHC = immunohistochemistry; ER = estrogen receptors; PR = progesterone receptors; EMA = epithelial membrane antigen; SMA = alpha smooth muscle actin; R0 = negative margins; R1 = positive margins; VB = vaginal birth; CS = cesarean section; NED = no evidence of disease; RV = radical vulvectomy; LE = local excision; REL = relapse, NA = not available, PFS = progression-free survival.

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
