# Peer review of "Aggressive Angiomyxoma of the Lower Female Genital Tract in Pregnancy: A Review of the MITO Rare Tumors Group"

_cancers, 2023, doi:10.3390/cancers15133403_

Round 1
Reviewer 1 Report
Please find my comments and suggestions below.
(1) In the Discussion section, what does this manuscript contribute to? The authors are encouraged to conduct their own assessment and include a dedicated section that outlines future scenarios of aggressive angiomyxoma.
(2) In the Conclusion section, state the most important result. I would recommend that the authors simplify their conclusions and provide a concise summary of the main results.
(3) In the Abstract section, please consider further simplifying and summarizing the main conclusions of aggressive angiomyxoma to enhance readability.
(4) Table1, I suggest changing the ‘ID’ to ‘Reference’; ‘location’ to ‘Location’.
(5) Please check for abbreviations.
Author Response
- In the Discussion section, what does this manuscript contribute to? The authors are encouraged to conduct their own assessment and include a dedicated section that outlines future scenarios of aggressive angiomyxoma.
Ok, done
- In the Conclusion section, state the most important result. I would recommend that the authors simplify their conclusions and provide a concise summary of the main results.
Ok, done
- In the Abstract section, please consider further simplifying and summarizing the main conclusions of aggressive angiomyxoma to enhance readability.
Ok, done.
- Table1, I suggest changing the ‘ID’ to ‘Reference’; ‘location’ to ‘Location’.
Ok, done.
- Please check for abbreviations.
Ok, done.
Thank you for your revision.
Reviewer 2 Report
This is a very difficult type of paper to review. With such a rare and obscure tumour most of us have never seen a case. This review found only 13 appropriate cases.
All i can say is that this is useful in terms of highlighting a rare condition but adds little else to the literature. It is really no possible to add much.
The authors would seem to provide some guidance on management but with super rare conditions and very heterogeneous collection it is difficult to draw many conclusions. A few minor grammatical comments below
Some odd words which are not typical of English language but probably reflect translation.
Line 18 in summary is not grammatically correct.
Line 229 demolitive is a neologism to me
Author Response
This is a very difficult type of paper to review. With such a rare and obscure tumour most of us have never seen a case. This review found only 13 appropriate cases.
All i can say is that this is useful in terms of highlighting a rare condition but adds little else to the literature. It is really no possible to add much.
The authors would seem to provide some guidance on management but with super rare conditions and very heterogeneous collection it is difficult to draw many conclusions. A few minor grammatical comments below
Comments on the Quality of English Language
Some odd words which are not typical of English language but probably reflect translation.
Line 18 in summary is not grammatically correct.
Ok, we have corrected it.
Line 229 demolitive is a neologism to me
Ok, we have corrected it.
Thank you for revisions.